# Multi-objective maintenance optimization of contact networks considering integrated state parameters

Zhijian Qu[1,2], Xinyu Liao[1,2]*, Rui Chi[2], Baoquan Wei[2], Xinxing Hou[1]

1 State Key Laboratory of Performance Monitoring and Protecting of Rail Transit Infrastructure, East China Jiaotong University, Nanchang, Jiangxi Province, China, 2 Electrical and Automation Engineering Collage, East China Jiaotong University, Nanchang, Jiangxi Province, China

* 2022028085801015@ecjtu.edu.cn

## Abstract

With the gradual expansion of China's high-speed railroad operation scale and the accumulation of operation time, the pressure of operation and maintenance of contact networks is increasing, and the operation and maintenance strategy of railroad contact networks under the traditional cyclic maintenance mode has the double dilemma of low maintenance efficiency and high labor intensity of maintenance personnel. In order to protect the transportation quality and efficiency of high-speed railroads and reduce the labor intensity of maintenance personnel, this paper proposes a multi-objective optimization operation and maintenance strategy for contact networks considering integrated state parameters, which is able to ensure the high reliability of contact networks while reducing the cost of maintenance and reducing the man-hours required for maintenance. Firstly, according to the integrated state parameters such as the guide height, contact force, pull-out value, and remaining useful life (RUL) of contact networks, the state of contact networks near specific towers is analyzed, revealing the initial reliability of the main equipment including contact wires, positioners, hanging strings, etc., which is used as a parameter of the multi-objective optimal maintenance model. Then, a multi-objective optimal maintenance model is established to achieve the optimization goal of high average reliability of contact networks, low maintenance cost and short required working hours. Finally, the model is solved by the multi-objective optimization algorithm to obtain a series of maintenance solutions in the form of Pareto non-inferior solution sets. Simulation calculations with data from a railroad bureau in southern China yielded that the average reliability of the contact networks system was 0.969, and the maintenance cost was 1,029 thousand yuan, spending 71 man-hours, when the maintenance method generated by the operation and maintenance strategy mentioned in this paper was used. Compared with the traditional maintenance method, the maintenance scheme obtained by the proposed in this paper optimizes the required man-hours and improves the labor productivity while maintaining higher reliability and lowering the maintenance cost.

**Data availability statement:** All relevant data are within the paper and its Supporting Information files.

**Funding:** This work was supported by the State Key Laboratory of Performance Monitoring and Protecting of Rail Transit Infrastructure (HJGZ2022203). This research was also supported by the National Natural Science Foundation of China (52462052), Jiangxi Provincial Natural Science Foundation Key Project Fund (20232ACB204025), and Jiangxi Provincial High-level and High-skilled Leading Talents Cultivation Project Fund (202223323). There was no additional external funding received for this study. The funders had no role in study design, data collection and analysis, decision to publish, or preparation of the manuscript.

**Competing interests:** The authors have declared that no competing interests exist.

## 1. Introduction

A contact network acts as an important part of the infrastructure of a railroad power supply. Its reliability in high-speed railroad is directly related to the safety of travel. When the contact network is in operation, defects and repairs occur repeatedly due to a variety of factors and are accompanied by high maintenance costs. For a long time, in the operation and maintenance of a contact network, there is a contradiction that "high performance of the equipment is expected to operate while maintenance resources are limited" [1]. At this stage, the main maintenance mode in China is not precise enough, and is prone to "under-maintenance" and "over-maintenance". Therefore, the more accurately to overhaul the equipment, there is a strong practical need to determine the current state of contact networks according to the inspection data thereof, which is also a basis for guiding the arrangement of the maintenance plan of contact networks. At the same time, due to the limited time for maintenance and the multiplicity of maintenance tasks, and the contact network being suspended in the air, resulting in high labor intensity of maintenance personnel and high-risk factor of overhaul. Therefore, the study of a multi-objective optimized maintenance strategy for contact networks based on integrated state parameters detection, and the arrangement of maintenance plan according to the actual state of the contact networks can effectively improve the reliability of the contact network system, reduce the maintenance cost, reduce the man-hours required for the maintenance, improve the labor productivity, and realize the accurate maintenance.

In recent years, scholars have investigated the optimization of maintenance decisions of contact networks. Some scholars assessed the state of a contact network [2] using a Hidden Markov Model (HMM) to simulate the degradation of a high-speed rail contact network; the identification accuracy of the resulting model reached about 90%, but the HMM model can be used only to analyze the discrete observed quantities, and the integrated state detection parameters such as the guide height and contact force are continuous, and part of the information will be lost by discretization. Others [3] modeled the assessment of the degradation state of contact networks based on dynamic and static measured parameters using the hidden Markov model with a mixed Gaussian probability density function (CHMM), which is capable of analyzing and identifying the continuous state parameters of contact networks, but the parameters used only include the guide height and contact force, which is not comprehensive enough. Elsewhere, other workers [4] utilized the Hilbert-Huang transform (HHT) method to decompose a series of contact pressure, pull-out value, guide height, and hard-point data signals, and obtained the correlation and degradation trend between the four parameters, but since HHT is essentially a signal decomposition method, it cannot be used to identify the state of the parameters, and it is difficult to direct maintenance based on the state of contact networks. Overall, although the aforementioned studies have achieved some results, they have not yet gone so far as to use the identified states to guide the arrangement of maintenance plans.

Some scholars also studied how to predict the RUL of contact networks. Previous research [5] presented a deep neural network-based CCS-DNN model to anticipate the RUL of contact network, but did not utilize the remaining useful life (RUL) to direct the development of maintenance plans. Others [6] proposed a health management model with RUL for use in the assessment of rolling bearings, and based on the state and RUL, the rolling bearing maintenance strategy was studied, but its maintenance strategy used a game model, which could not respond well to the relationship between maintenance cost and the reliability of the contact network. Overall, although these studies achieved some progress, they do not make good use of RUL to guide the maintenance strategy.

Some researchers have explored the maintenance optimization model of the contact network system: in [7], a mathematical model is established for reliability analysis of key

equipment in a traction power supply system on the basis of the Weibull distribution model, and the failure time of the parts of this method is Weibull distributed, but it is assumed that the failed parts are irreparable and the replacement of the parts does not affect the reliability. Based on the theory of delay time, others [8] established a subsystem multi-level optimization model for the maintenance interval and the system into a group maintenance optimization model, and used rolling time domain and dynamic programming methods to solve the model, thus reducing the maintenance downtime and saving the maintenance cost. The development of maintenance plans has a certain guiding significance, but the method does not take the degree of reliability as an optimization objective, the resulting maintenance plan corresponds to the unknown degree of reliability. In [9], a maintenance decision objective of reducing maintenance cost and the number of unanticipated failures was mooted, but the method must strike a balance between economy and reliability. Elsewhere [10] developed a multi-objective optimization model for contact networks that can simultaneously consider reliability and maintenance cost targets, which can ensure the highest possible system reliability while reducing maintenance costs, but does not take into account the effects of the current state of the equipment on the degree of reliability. A maintenance model with the objectives of reliability and maintenance cost was established [11] to optimize the system with the improved NSGA2 algorithm, which further optimizes the reliability and maintenance cost of the resulting maintenance strategy, but does not consider the optimization of the required man-hours. In [12], researchers studied the maintenance staffing scheme and proposed a preventive maintenance staffing optimization management system for contact networks, and in [13] a prediction model for the demand of maintenance man-hours for use with weapons and equipment repair and maintenance was proposed, but the combination of reliability and maintenance cost was not optimized. In general, most of current research is still based on cycle repair, without considering the current state of the equipment, and workers are yet to explore the optimization of the maintenance hours required, so that there is a certain degree of blindness in its application in practice.

Some scholars have researched on maintenance optimization algorithms for contact network systems. In the field of multi-objective maintenance decision optimization of contact networks, the NSGA-II optimization algorithm is mainly applied at present. However, Deb [14] suggested that the NSGA-II algorithm has poor convergence and diversity when solving three or more objective optimization problems. In research areas such as bridge network maintenance optimization [15] and path optimization [16], NSGA-III has been employed in optimization problems with three and more objectives, and the convergence accuracy and diversity of the resulting solutions are better than that of NSGA-II, however, in the maintenance optimization of contact networks, there are few reports related to NSGA-III, and the present research is deemed urgent.

In summary, there are the following gaps in past research on contact networks maintenance optimization: (1) past research usually only focuses on single-objective or bi-objective optimization for reliability and maintenance cost, while the multi-objective optimization research on maintenance hours is in a blank state; (2) past research mainly focuses on how to predict the remaining service life, but research on how to effectively use this information to optimize the contact network maintenance program is rare; (3) the default of past research is to design maintenance programs for contact networks with an initial reliability of 1, i.e., they cannot be applied to contact networks that have just been put into service. The default initial reliability of contact networks is 1, i.e., the maintenance programs are designed for contact networks that have just been put into service, and cannot be applied to contact networks that have been put into service.

Aiming at the above problems, a multi-objective optimal operation and maintenance strategy for a contact network considering integrated state parameters is proposed. Firstly, through analysis of the contact force, guide height, pull-out value, and other integrated state parameters near the contact network pillars, the corresponding initial reliability of seven main items of equipment comprising the contact network is determined as the parameters of the multi-objective optimal maintenance model. Then, a multi-objective optimal maintenance model is established with the average reliability, maintenance cost and required working hours of the contact network as the optimization objectives. Finally, the multi-objective optimization algorithm is adopted to solve the model, so that the three objectives of average reliability, maintenance cost and required working hours of the contact networks can be optimized at the same time, and a series of Pareto non-inferior solution sets are obtained. This method ensures the reliability of system power supply, realizes the precise configuration and dynamic optimization of maintenance resources, effectively avoids the safety risk of "under-maintenance" and the waste of resources of "over-maintenance" that exist in the traditional maintenance mode, improves labor productivity, and provides key technical support for the construction of intelligent operation and maintenance system of contact networks.

## 2. Identification of the state of contact networks and determination of the initial reliability

### 2.1 State identification of integrated state parameters of the contact networks

Parameters such as guide height and pull-out value were evaluated using the index rating table for integrated state parameters of contact networks to obtain the rating and status of each parameter. The obtained score ranges from 0.1 to 1, and is divided into five grades: excellent, good, medium, poor, and very poor according to the score. According to the literature [17,18], the index rating for integrated state parameters of contact networks is listed in Table 1.

After obtaining the score and status of each parameter, according to the contact network 1C inspection defect information data sourced from a railroad bureau in southern China, the numbers of times that the guide height, contact force, intra-span height difference, pull-out value, hard point, and network pressure are sub-standard caused by the contact wire, bearing cable, insulator, central anchor joint, compensator, locator, and hanging string are counted. In this way, the mapping relationship between each parameter and each equipment is obtained,

**Table 1. Index rating for integrated state parameters of contact networks.**

| State | Excellent | | Good | | Medium | | Poor | | Very poor | |
|---|---|---|---|---|---|---|---|---|---|---|
| Score | 1 | 0.9 | 0.8 | 0.7 | 0.6 | 0.5 | 0.4 | 0.3 | 0.2 | 0.1 |
| Guide height/mm | (5970, 6030) | 6000 ± 30 | (6030, 6050) or (5950, 5970) | 6000 ± 50 | (6050, 6100) or (5900, 5950) | 6000 ± 100 | (6100, 6150) or (5850, 5900) | 6000 ± 150 | (6150, 6200) or (5800, 5850) | ≥6200 or ≤5800 |
| Contact force/N | 120 | (120, 150) or (90, 120) | 120 ± 30 | (150, 180) or (60, 90) | 120 ± 60 | (180, 200) or (40, 60) | 120 ± 80 | (200, 220) or (20, 40) | 120 ± 100 | >220 or <20 |
| In-span height difference/mm | 0 | (0, 25) | 25 | (25, 50) | 50 | (50, 100) | 100 | (100, 150) | 150 | >150 |
| Pull-out value/mm (absolute value) | [0, 300] | (300, 330) | 330 | (330, 360) | 360 | (360, 400) | 400 | (400, 450) | 450 | >450 |
| Hard point/N | [0, 20] | (20, 30) | 30 | (30, 50) | 50 | (50, 100) | 100 | (100, 150) | 150 | >150 |
| Contact voltage/kV | (23.5, 26.5) | 25 ± 1.5 | (26, 27) or (22, 23) | 25 ± 2 | (27, 27.5) or (22.5, 23) | 25 ± 2.5 | (27.5, 28) or (22, 22.5) | 25 ± 3 | (28, 29) or (21, 22) | ≥ 29 or ≤ 21 |

and the score and status of each parameter are transformed into an equipment score and status through the mapping relationship into the rating and status of the equipment.

The defect inspection data of the contact network 1C used in the present research were sourced from a railroad bureau in southern China, and the data were collected from December 2019 to December 2020. A total of 231 valid defect data were obtained, mainly arising from hanging chords and positioners, including 136 guide height defects, 55 intra-span height difference defects, 38 contact force defects, and one pull-out value defect.

Some 59.13% of the guide height defects were caused by hanging chords; 23.91% of the in-span height difference defects; 16.96% of the contact force defects; and 0% of the pull-out value defects.

Some 40% of guide height defects were caused by positioners; 13.33% of in-span height difference defects; 40% of contact force defects; 6.67% of pull-out value defects.

Let the state value of the guide height be $X$, the state value of the height difference within the span be $Y$, the state value of the contact force be $Z$, and the state value of the pull-out value be $W$. Letting the state value of the lifting chord be $M$ and the state value of the positioner be $N$, then the mapping relation can be written as:

$$M = 0.5913X + 0.2391Y + 0.1696Z \tag{1}$$

$$N = 0.4X + 0.1333Y + 0.4Z + 0.067W \tag{2}$$

If the state value is between [0.9, 1], the state is set to excellent; if the state value is between [0.8, 0.9), the state is set to good; if the state value is between [0.7, 0.8), the state is set to medium; if the state value is between [0.6, 0.7), the state is set to poor; and if the state value is between [0, 0.6), the state is set to very poor.

## 2.2 Determination of initial reliability of main equipment of contact networks

After determining the state value and state level of each device, the state of the device is transformed into the initial reliability of the device based on the relationship between the state and the initial reliability. According to [6], there are three cases for the value of initial reliability, as listed in Table 2.

In the range of reliability, the following three cases are separately considered to explore the value of initial reliability, and comparative tests are performed in the arithmetic case analysis to determine the way of taking the value of initial reliability: ① $R$ = [0.95, 0.75, 0.5, 0.25, 0.05]; ② $R$ = [0.9, 0.6, 0.4, 0.1, 0.01]; ③ $R$ = [0.99, 0.85, 0.59, 0.35, 0.09].

After determining the initial reliability of the seven major devices, they are input in the multi-objective optimal maintenance strategy model along with the RUL values as the parameters of the model.

**Table 2. Correspondence between equipment state and initial reliability.**

| State | Initial reliability range | Retrieve value |
|---|---|---|
| Excellent | R ∈ [0.9, 1] | $R = 0.95$; $R = 0.9$; $R = 0.99$ |
| Good | R ∈ [0.6, 0.9) | $R = 0.75$; $R = 0.6$; $R = 0.85$ |
| Medium | R ∈ [0.4, 0.6) | $R = 0.5$; $R = 0.4$; $R = 0.59$ |
| Poor | R ∈ [0.1, 0.4) | $R = 0.25$; $R = 0.1$; $R = 0.35$ |
| Very Poor | R ∈ [0, 0.1) | $R = 0.05$; $R = 0.01$; $R = 0.09$ |

## 3. Modeling of multi-objective optimal maintenance strategies

A multi-objective optimal maintenance strategy model is established taking the maintenance scheme, the initial reliability and RUL of the seven main devices of the contact network as inputs, and the reliability of the contact network, the total maintenance cost and the required man-hours as outputs.

The maintenance program assesses how the seven major items of equipment such as contact wires, bearing cables, *etc.* are maintained within the RUL. The input maintenance program is a table with seven rows indicating the seven major items of equipment; the number of columns represents the number of maintenance phases, each maintenance phase lasts for 15 days, as many as there are 15 days in the RUL, matching the number of maintenance phases there are. The maintenance mode is: no repair, minor repair, major repair, or replacement, a total of four kinds of maintenance; the form of the input maintenance program is shown in Fig 1.

The reliability of major equipment $n$ at stage $k$ as a function of time is expressed by the Weibull function as $R_{n,\,k}(t)$.

$$R_{n,\,k}(t) = R_{n,\,0,\,k} * e^{-\left(\frac{t}{m_j \alpha_n}\right)^{\beta_n}} \tag{3}$$

In (3) $n$ denotes major equipment, there are seven such major items of equipment, so $n = 1,\,2,\,\ldots\ldots,\,7$; $k$ represents equipment in the $k^{\text{th}}$ maintenance stage, $k = 1,\,2,\,\ldots\ldots,\,N_p$; $N_p$ is the number of maintenance stages within RUL, *i.e.*, the number of columns in the maintenance program.

$R_{n,\,0,\,k}$ denotes the initial reliability of major equipment $n$ in stage $k$; in particular, $R_{n,\,0,\,1}$ is the initial reliability of the major device $n$ in stage 1.

$\alpha_n$ and $\beta_n$ represent the scale parameter and shape parameter of the major equipment $n$ in Weibull function, respectively; $j$ is the maintenance mode, and $j = 1,\,2,\,3,\,4$ represents

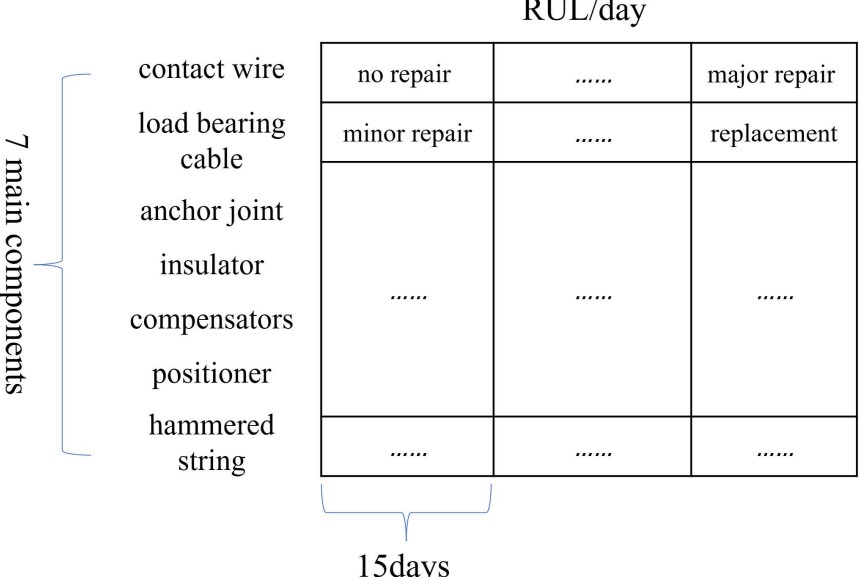

**Fig 1. Design of maintenance programs.**

no repair, minor repair, major repair, and replacement, respectively; $m_j$ is the improvement factor when adopting repair mode $j$, where: $m_0 = 1$; $m_1 = 0.8$; $m_2 = 0.8$; $m_4 = 1$;

The reliability of all but seven major devices is represented by an exponential function as $R_c(t)$:

$$R_c(t) = e^{-\gamma t} \tag{4}$$

$\gamma$ denotes the argument of the exponential function, $\gamma > 0$, $t > 0$;

If no repair or minor repair is utilized at the beginning of the stage $k$, $R_{n,\,k}(t)$ in $R_{n,\,0,\,k}$ can be expressed as:

$$R_{n,\,0,\,k} = R_{n,\,f,\,k-1} \tag{5}$$

where $R_{n,\,f,\,k-1}$ is the reliability of the item of major equipment $n$ at the end of the maintenance stage $k-1$.

If the maintenance mode is used at the beginning of stage $k$, $R_{n,\,k}(t)$ in $R_{n,\,0,\,k}$ can be expressed as follows:

$$R_{n,\,0,\,k} = R_{n,\,f,\,k-1} + m_2 \left( R_0 - R_{n,\,f,\,k-1} \right) \tag{6}$$

where $R_0$ is the starting reliability of the new component, and $R_0 = 1$.

If the start of the stage $k$ is repaired using complete replacement, $R_{n,\,k}(t)$ in $R_{n,\,0,\,k}$ can be expressed as:

$$R_{n,\,0,\,k} = R_0 \tag{7}$$

$R_{n,\,f,\,k-1}$, the reliability of the major equipment $n$ at the end of the $k$-1$^\text{th}$ maintenance stage $k$-1 can be expressed as:

$$R_{n,\,f,\,k-1} = R_{n,\,k-1}(t_p) \tag{8}$$

where $t_p$ denotes the time for each maintenance stage, and each maintenance stage is 15 days, or half a month, so $t_p = 0.5$;

The probability that a contact network will operate properly is the product of the probability that each piece of equipment will operate properly. The seven main devices are independent of each other, and the other parts of the contact network system are also mutually independent. The whole contact network system can be in a state of normal operation and the probability thereof can be expressed thus:

$$R_{dt,\,k}(t) = R_c(t) \prod_{n=1}^{7} \left\{ 1 - \sigma_n \left[ 1 - R_{n,\,k}(t) \right] \right\} \tag{9}$$

where $\sigma_n$ denotes the probability that when item of equipment $n$ fails to operate properly making the overall system fail to operate properly. The average probability that the system can operate normally is given by:

$$R_{avg} = \frac{\sum_{k=1}^{N_p} \frac{1}{t_p} \left( \int_{(k-1)t_p}^{kt_p} R_{dt,\,k}(t)dt \right)}{N_p} \tag{10}$$

The total maintenance cost function is as follows:

$$C_{sys} = \sum_{n=1}^{7} \left[ \sum_{j=1}^{4} C_{n,\,j} \left( \sum_{k=1}^{N_p} P_{n,\,k,\,j} \right) \right] + \sum_{n=1}^{7} \left( \left( C_{n,\,h} \sum_{k=1}^{N_p} \int_{(k-1)t_p}^{kt_p} h_{n,\,k}(t)dt \right) \right) \tag{11}$$

where $C_{n,j}$ represents the cost of taking the maintenance mode $j$ for the item of major equipment $n$ in question; $P_{n,k,j}$ represents whether the major equipment $n$ takes the maintenance mode $j$ in maintenance stage $k$ or not, if it is yes then $P_{n,k,j} = 1$, if not, then $P_{n,k,j} = 0$; $C_{n,h}$ represents the cost of the emergency repair cost when the equipment $n$ is not operating normally, $h_{n,k}(t)$ represents the probability that the equipment $n$ cannot operate normally in the maintenance stage $k$, and $N_p$ represents the number of maintenance stages.

$h_{n,k}(t)$ can be denoted thus:

$$h_{n,k}(t) = -\frac{1}{R_{n,k}(t)}\frac{dR_{n,k}(t)}{dt}, \ (k-1)t_p \le \ t \ \le kt_p \tag{12}$$

where $R_{n,k}(t)$ represents the probability that item of equipment $n$ will function properly in maintenance stage $k$.

The total man-hours required are as follows:

$$MH = \sum_{n=1}^{7}\sum_{k=1}^{N_p}\sum_{j=1}^{4}\left(P_{n,k,j} * MH_{n,j}\right) \tag{13}$$

where $MH_{n,j}$ is the number of hours required for the major equipment $n$ to take maintenance mode $j$.

The three functions of Eq. (10) giving the average reliability of contact network system $R_{avg}$, Eq. (11) giving the maintenance cost of the contact network system $C_{sys}$, and Eq. (13) giving the required man-hours $MH$ are adopted as the main functions of multi-objective optimization, which are optimized by the improved optimization algorithm to determine the optimized maintenance strategies for the three indicators at the same time.

## 4. Solving the multi-objective optimal maintenance model

A multi-objective optimization algorithm is used for optimizing $R_{avg}$, $C_{sys}$, and $MH$, so that the average system reliability is as high as possible, the maintenance cost and required man-hours are as low as possible, and the corresponding maintenance solutions are obtained. The optimization result is a subset of the set of Pareto optimal solutions. The schematic diagram of the multi-objective optimization algorithm is displayed in Fig 2.

When coding the maintenance programs in the maintenance model of contact networks, each device in the maintenance scheme corresponds to a repair mode at each maintenance stage. There are a total of four repair methods, which are represented by two Boolean numbers, 00 for no repair, 01 for minor repair, 10 for major repair, and 11 for complete replacement. There are seven types of equipment in the repair program, and the number of repair stages is $N_p$, so the length of the code is $7 \times N_p \times 2$.

Secondly, the initial population of maintenance programs is generated and utilized as a new population, which contains 100 individuals. A multi-objective optimization model involving 100 maintenance programs is established to obtain the average reliability, maintenance cost of the system, and required man-hours corresponding to each maintenance program. Then, whether or not the number of iterations is reached is determined. If the number of iterations is not reached, the maintenance programs are ranked and divided into several grades according to the evaluation of system average reliability, maintenance cost and required man-hours. The top-ranked individuals in each rank are taken to perform crossover, mutation and other operations to form a new population; finally, if the number of iterations is reached, these maintenance programs and their corresponding average system reliability, maintenance cost, and required man-hours are output in the form of a Pareto optimal solution set.

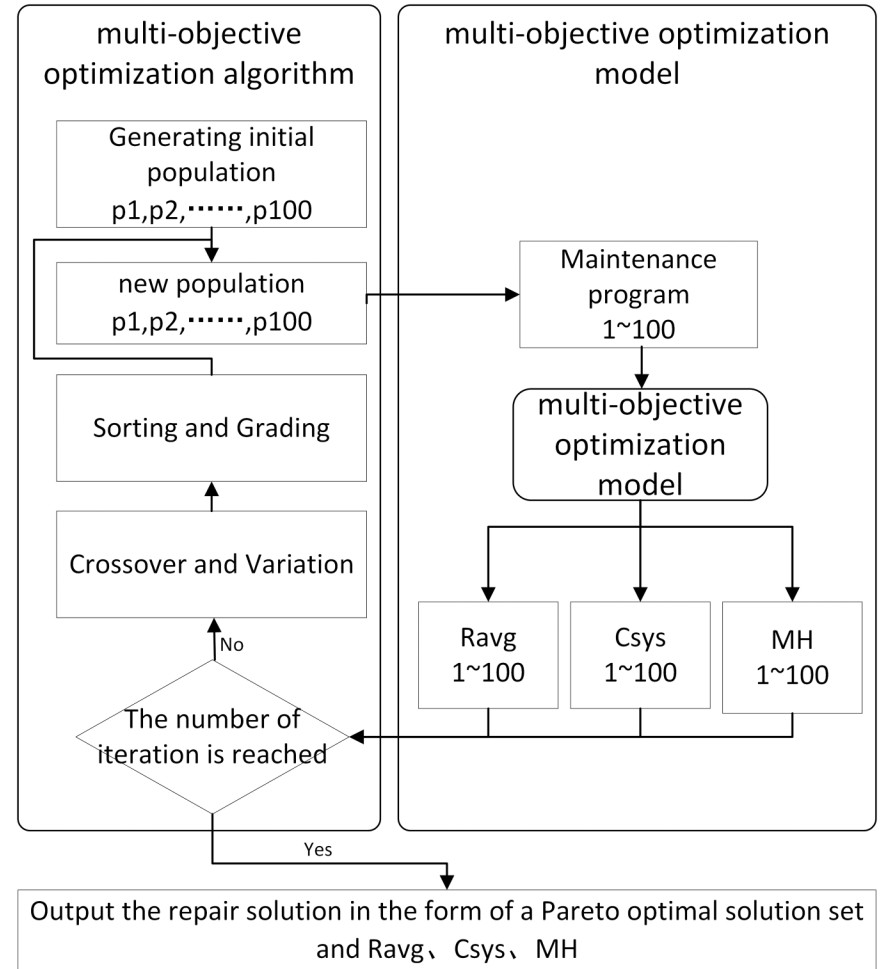

**Fig 2. Schematic diagram of the multi-objective optimization algorithm.**

For example, a population has four individuals, each corresponding to a maintenance program. First, the individuals are sorted according to the fitness value (*i.e.*, average reliability, repair cost, and required man-hours) and the non-dominated ranking of each individual, and the sorted order is: Individual 3, Individual 1, Individual 2, and Individual 4. The crossover and mutation operations are then performed to generate the new offspring individuals and obtain the next generation of the population: *e.g.*, Individual 1 crosses over with Individual 2 to obtain the offspring Individual 5; Individual 3 crosses over with Individual 4 to obtain the offspring Individual 6. The offspring individuals are mutated: the offspring Individual 5 is mutated to get the offspring Individual 7; the offspring Individual 6 is mutated to get the offspring Individual 8. Finally, according to the calculated value of fitness and non-dominated ranking, the next generation of populations are filtered to get the next generation of the population: Individual 3, Individual 1, Individual 7, and Individual 8, and one iteration is completed to reach the maximum number of iterations to output the optimization scheme.

## 5. Case study

Data about the contact networks near a tower on a line of a railroad bureau in southern China are used as an example for analysis. Firstly, the parameter state is identified, the initial

reliability of the equipment is determined, and then the reliability and RUL of the equipment are input into a multi-objective optimal maintenance model, and then the multi-objective optimization algorithm solves the model to obtain the maintenance program that optimizes the system average reliability, maintenance cost, and required man-hours simultaneously. The inspection data of contact networks in the vicinity of a pole tower can be expressed as a vector: $A = [a_1, \ b_1, \ c_1, \ d_1, \ e_1, \ f_1, \ g_1]^{\mathrm{T}}$, where $a_1$ is the guide height data, $b_1$ is the hard point data, $c_1$ is the contact force data, $d_1$ is the web pressure data, $e_1$ is the pull-out value data, $f_1$ is the in-span height difference data, and $g_1$ is the RUL value backward derived from the defective record table. $a_1 = 5986$; $b_1 = 2$; $c_1 = 159$; $d_1 = 24566$; $e_1 = -225$; $f_1 = 45$; $g_1 = 110$.

According to the criteria listed in Table 1, the state values of the guide height, height difference within the span, contact force, and pull-out value are $X = 1$, $Y = 0.7$, $Z = 0.7$, and $W=1$, respectively. The state value of the lifting chord $M=0.8774$ and the state value of the positioner $N = 0.8393$ are obtained from Eqs. 1 and 2. The state value of the lifting chord and the positioner can be set as good, while the state value of the other items of equipment is deemed excellent. The RUL is 110 days, so if each maintenance interval is 15 days, there are eight maintenance phases.

## 5.1 Maintenance model parameters

The data of contact networks related components for a section of the line in [10] are listed in Table 3.

where $\alpha_n$ denotes the scale parameter in the Weibull function of equipment $n$, $\beta_n$ denotes the shape parameter in the Weibull function pertaining to the item of equipment $n$; $\sigma_n$ is the probability that the failure of the equipment $n$ in the contact network system results in the failure of the whole system; $C_1$ denotes the repair cost required for minor repairs, $C_2$ denotes the repair cost required for major repairs, $C_3$ denotes the repair cost required for complete replacements, $C_h$ indicates the cost of *ex post facto* emergency repairs.

After on-site research in the Railway Power Supply Work Area: ① The number of workers required for each equipment unit in the case of minor repairs, major repairs and complete replacement is about 12. ② The time spent on minor repairs for all seven major items of equipment in Table 3 is about 10 minutes; the time spent on major repairs (Table 3) is about 30 minutes; the time required for the complete replacement of contact wires and bearing cables is about 10 hours, the time required for the complete replacement of central anchor joints and locators is about 30 minutes, the time required for the complete replacement of insulators is about 20 minutes, the time required for the complete replacement of the compensators is approximately 1 hour and insulators, with an additional 15 minutes for complete replacement of hanging strings.

**Table 3. Parameters and maintenance costs associated with each piece of equipment in a section of the contact network (maintenance cost/1, 000 ¥ ).**

| Equipment | $\alpha_n$ | $\beta_n$ | $\sigma_n$ | $C_1$ | $C_2$ | $C_3$ | $C_h$ |
|---|---|---|---|---|---|---|---|
| Contact wire | 143.47 | 4.55 | 1.0 | 10 | 100 | 1000 | 300 |
| Bearing cable | 171.32 | 21.12 | 1.0 | 10 | 100 | 1000 | 300 |
| Insulator | 179.21 | 5.25 | 1.0 | 20 | 60 | 500 | 120 |
| Central anchor joint | 175.18 | 9.14 | 1.0 | 2 | 8 | 80 | 16 |
| Compensator | 171.62 | 6.57 | 1.0 | 5 | 30 | 120 | 50 |
| Locator | 91.5 | 2.5 | 0.8 | 8 | 20 | 40 | 60 |
| Hanging string | 85.5 | 2.3 | 0.5 | 50 | 100 | 300 | 450 |

The number of workers required is multiplied with the time required to obtain the number of man-hours required.

## 5.2 Determination of initial reliability values

To determine the value of the initial reliability, 50 items of data pertaining to a line in a railroad bureau in southern China are randomly selected and assessed using three different reliability values. When the upward value is adopted, 75% of the data have the highest reliability of 0.95 or more; when the centered value is adopted, 48% of the data have the highest reliability of 0.95 or more; and when the downward value is used, no data have the highest reliability of 0.95 or more. According to [19], the reliability of the contact network system reaching 0.95 can be regarded as satisfying the operational safety requirements. To ensure that most of the data can meet the reliability safety requirements, the upward value is adopted. In the data taken in this experiment, the hanging string and locator are in a good status, and the status of other equipment is excellent, so the initial reliability of each equipment can be obtained as $R =$ [0.99, 0.99, 0.99, 0.99, 0.99, 0.85, 0.85].

## 5.3 Comparison of experimental results and algorithms

Taking NSGA-III multi-objective optimization algorithm as an example, the multi-objective optimal maintenance strategy model proposed herein is solved; the Pareto optimal solution set obtained is shown in Fig 3:

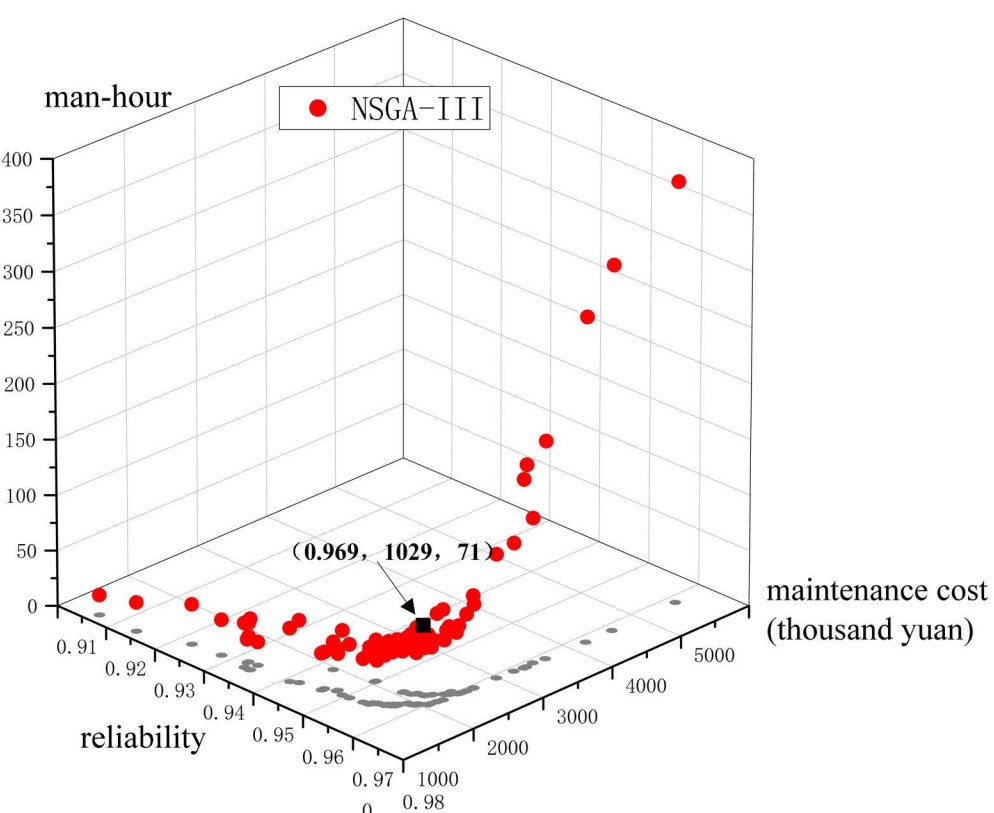

**Fig 3. Distribution of the Pareto frontier solution set.**

As shown in Fig 3, the red points are the Pareto frontier solution set and the gray points are the projection of the solution set on the reliability-maintenance cost plane. The reliability is distributed in the interval [0.916, 0.971], the repair cost is distributed between 154 and 4524 thousand dollars, and the number of required man-hours is between 18 and 377. From the distribution of the Pareto frontier solution set, it can be seen that, as the reliability is improved, the maintenance cost and the maintenance man-hours increase.

To verify the superiority of the multi-objective optimization strategy model for the maintenance of contact networks in the present research, the current mainstream algorithms within the field of multi-objective optimization for maintenance and the new algorithms proposed in recent years are compared. The algorithms are implemented in a multi-objective optimization platform in MATLAB [20]. The following configuration is used for the experiment: Microsoft Windows10, Intel® Core (TM) i7 -7700 CPU @ 3.60 GHz, with 8GB RAM. The population $N$ is set to 100 and the number of iterations is 100 generations.

The algorithms involved in the comparison are: NSGA-III, NSGA-II [21], gNSGA-II [22], TSNSGA-II [23], Adaw [24], DEAGNG [25], LMPFE [26], a total of seven algorithms. The results are shown in Fig 4.

1. Comparison of NSGA-III and NSGA-II (Fig 4a)

2. Comparison of NSGA-III and gNSGA-II (Fig 4b)

3. Comparison of NSGA-III and Adaw (Fig 4c)

4. Comparison of NSGA-III and DEAGNG (Fig 4d)

5. Comparison of NSGA-III and LMPFE (Fig 4e)

6. Comparison of NSGA-III and TSNSGA-II (Fig 4f)

As shown in Fig 4, the red points are the Pareto frontier solution sets for NSGA-III and the blue points are the Pareto frontier solution sets for the comparison algorithms. The result is such that the greater the reliability the better, and the lower the maintenance cost and required man-hours the better. All algorithms perform better in terms of reliability, with the greatest reliability reaching 0.971, but the performance in terms of maintenance cost and required man-hours varies more. When the reliability is certain, NSGA-III has lower maintenance cost and required man-hours compared with the other algorithms, and the front formed by NSGA-III is closer to the coordinate axes than when using the other algorithms, showing that the convergence accuracy of NSGA-III is better than the comparative algorithms.

Although the forms of solution set distributions of the various algorithms are similar, NSGA-III has a more centralized solution set distribution and better convergence. In terms of the objective value of each objective, compared with NSGA-II and other multi-objective optimization algorithms proposed in recent years, NSGA-III can obtain maintenance strategies with lower maintenance cost and fewer man-hours required, and the decision makers can be selected from a richer menu of choices while ensuring higher reliability of the contact network system.

The comparison of the results of each method at different reliability degrees is shown in Table 4:

At a reliability of 0.95, NSGA-III has the lowest maintenance cost of $446, 000. In terms of labor hours required, NSGA-III and LMPFE both have the lowest at 48 labor hours. At a reliability of 0.969, NSGA-III has the lowest maintenance cost and required labor hours, with a maintenance cost of $1, 029, 000 and 71 man-hours. At a reliability of 0.971, again NSGA-III has the lowest maintenance cost and required man-hours, with a maintenance cost of $2,

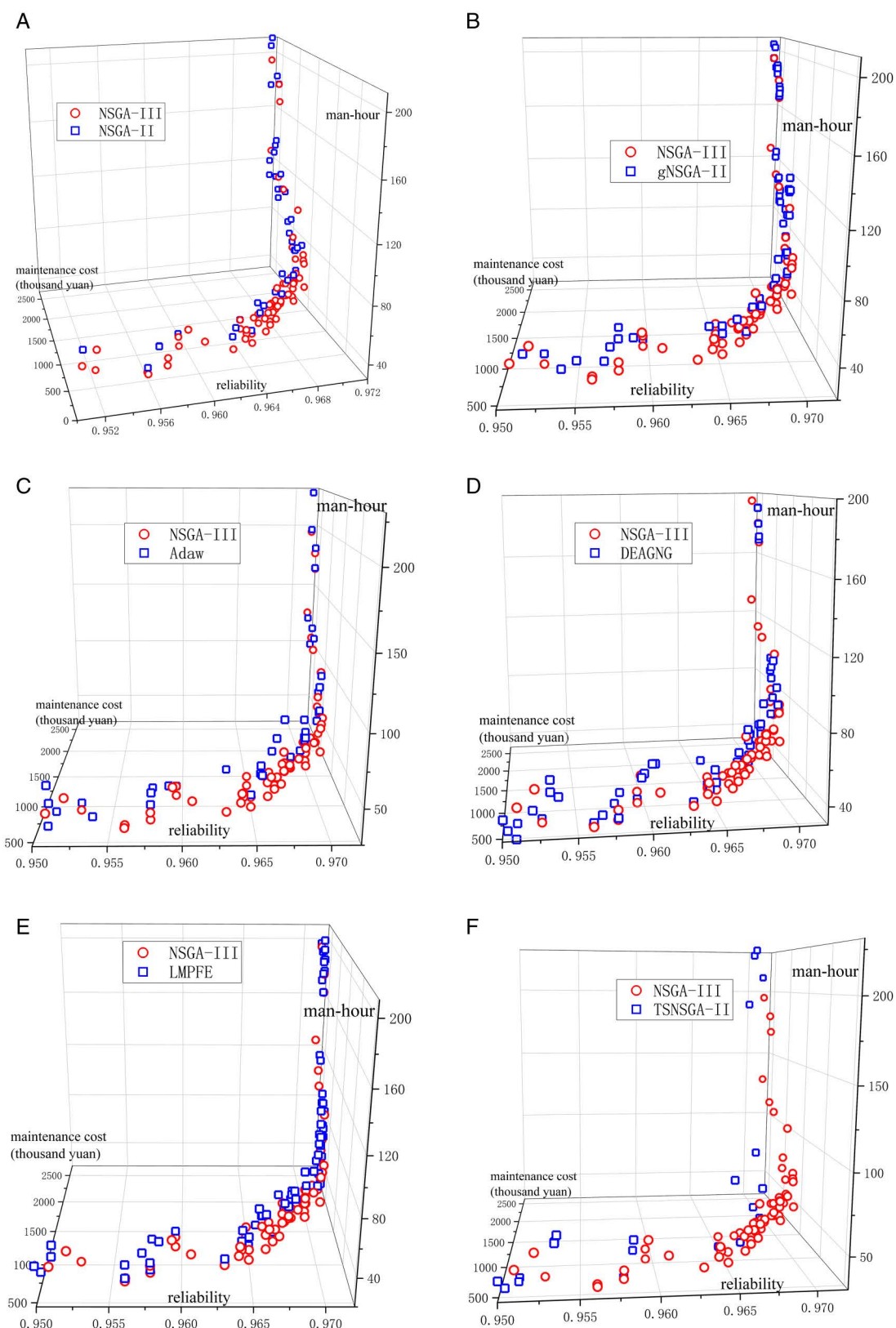

**Fig 4. Comparison of Pareto frontier solution sets for NSGA-III and other algorithms.**

**Table 4. Comparison of different algorithms.**

| | $R_{avg}$ 0.950 | | $R_{avg}$ 0.969 | | $R_{avg}$ 0.971 | |
| | $C_{sys}$(thousand ¥) | MH | $C_{sys}$(thousand ¥) | MH | $C_{sys}$(thousand ¥) | MH |
|---|---|---|---|---|---|---|
| LMPFE | 453 | **48** | 1153 | 79 | 2397 | 183 |
| NSGA-II | 528 | 56 | 1280 | 87 | 2389 | 187 |
| g-NSGA-II | 529 | 50 | 1176 | 77 | 2267 | 171 |
| TS-NSGA-II | 470 | 62 | 2194 | 231 | 2614 | 211 |
| Adaw | 545 | 50 | 1209 | 105 | 2452 | 126 |
| DEAGNG | 498 | 54 | 1086 | 73 | 2407 | 191 |
| NSGA-III | **446** | **48** | **1029** | **71** | **2218** | **115** |

218, 000 and 115 man-hours. From the data in Table 4, the results obtained by the NSGA-III algorithm are optimal.

The further to verify the superiority of the NSGA-III algorithm, NSGA-III is still adopted to solve the multi-objective maintenance strategy model for the contact network proposed in this paper and compared with the CSEA method used in the literature [27].

Taking a certain state value as the superior inspection data B of the contact network, expressed as: $B = [a_2, \ b_2, \ c_2, \ d_2, \ e_2, \ f_2, \ g_2]^T$, where $a_2 = 6001$; $b_2 = -1$; $c_2 = 144$; $d_2 = 25503$; $e_2 = -211$; $f_2 = 0$; $g_2 = 150$, comparison is made in two dimensions: reliability and maintenance cost. The Pareto optimal solution set obtained by following the above steps is shown in Fig 5.

The set of Pareto optimal solutions obtained from the inspection data B and the NSGA-III algorithm has a reliability distribution in the interval [0.970, 0.994] and a maintenance cost

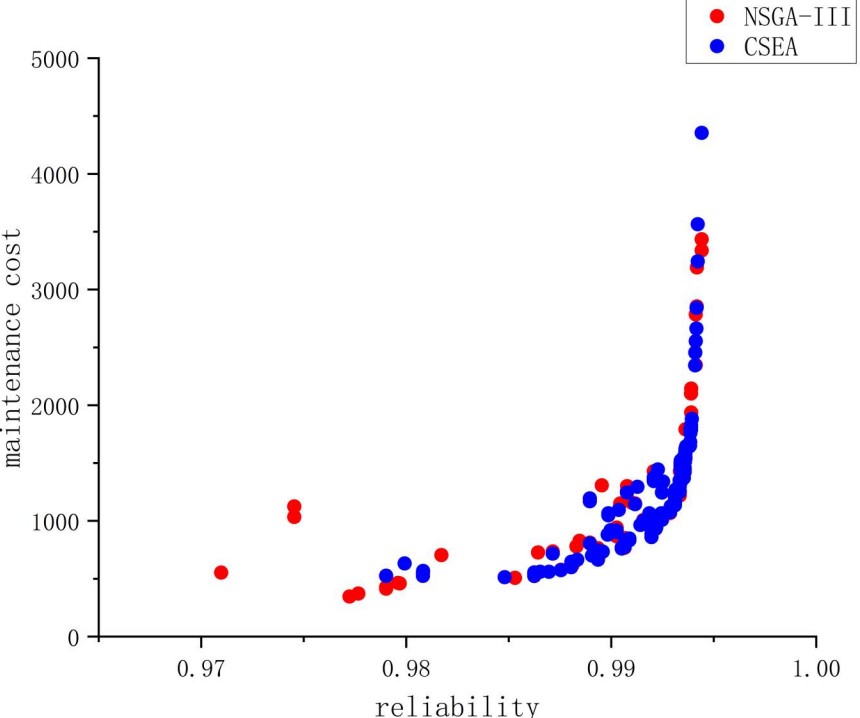

**Fig 5. Pareto solution set from test data B using two schemes.**

distribution between 346, 000 ￥ and 3, 435, 000 ￥. The CSEA methodology obtains a reliability distribution in the interval [0.979, 0.994] and a maintenance cost distribution between 525, 000 ￥ and 4, 355, 000 ￥.

From the Pareto front, the results of the proposed method are similar to those of the reference methods, but the CSEA method takes 1 hour and 36 minutes to run, while the proposed method takes only 10 seconds (the method described herein can also be used to optimize the required man-hours).

### 5.4 Maintenance programs received

The comparison results of different algorithms in Table 5 reveal that, as calculated from the inspection data A of the contact network, as its reliability is increased to greater than 0.969, the maintenance cost increases and any change in the number of required man-hours is insignificant in terms of the improvement of reliability, so the solution with the lowest maintenance cost and required man-hours is chosen as the maintenance program when the reliability is 0.969. As shown in Table 5, when the reliability is 0.969, the maintenance cost is 1, 029, 000 ￥, and the required man-hours are reduced by 71%.

The maintenance programs obtained from NSGA-III are shown in Table 5.

In Table 5, 1 means no repair, 2 denotes minor repair, 3 means major repair, and 4 means complete replacement. As can be seen from the maintenance program, the equipment was more heavily repaired with major repairs or complete replacements within the second and third maintenance phases, and only minor, or no, repairs were performed during the other maintenance phases.

This is due to the fact that the test data A used in this algorithm show the low initial reliability of all the equipment. In particular, the initial reliability of the locators and hanging strings is only 0.85, thus causing a relatively low initial reliability of the contact network system. Therefore, the hanging strings and the locators are replaced and the other equipment is overhauled during the first couple of maintenance cycles to improve the reliability of the contact network system. The reliability of the system is high after completing the maintenance, so only occasional minor repairs are required in the subsequent maintenance phases.

## 6. Conclusion

In order to improve the maintenance efficiency of contact networks and reduce the labor intensity of maintenance personnel, this paper proposes a multi-objective optimization operation and maintenance strategy for contact networks considering integrated state parameters. Firstly, the integrated state parameters of contact networks are analyzed to obtain the initial reliability of each main equipment of contact networks. Then, a multi-objective optimization maintenance model is established with the average reliability of contact

**Table 5. Maintenance program.**

| Equipment | Maintenance program | | | | | | | |
|---|---|---|---|---|---|---|---|---|
| Contact wire | 1 | 3 | 3 | 1 | 1 | 1 | 1 | 1 |
| Bearing cable | 1 | 3 | 3 | 1 | 1 | 1 | 1 | 1 |
| Insulator | 1 | 3 | 3 | 1 | 1 | 1 | 2 | 1 |
| Central anchor joint | 2 | 3 | 2 | 1 | 1 | 1 | 1 | 1 |
| Compensator | 1 | 3 | 3 | 1 | 1 | 1 | 2 | 1 |
| Locator | 1 | 4 | 1 | 1 | 1 | 1 | 1 | 1 |
| Hanging string | 1 | 4 | 1 | 1 | 1 | 1 | 1 | 1 |

networks, maintenance cost and required man-hours as the optimization objectives. Finally, the multi-objective optimization algorithm is used to solve the model and obtain a series of maintenance programs. The results show that this method arranges the maintenance plan according to the actual state of the equipment, which can improve the reliability of the contact networks, reduce the maintenance cost, reduce the maintenance man-hours, and provide effective guidance for the operation and maintenance work of the contact networks in the field. In this paper, the following conclusions are obtained in the analysis process:

(1) By analyzing the contact networks 1C detection defect information data of a railroad bureau in southern China, it is concluded that the status of contact networks hanging strings and locators is related to the guide height, the height difference within the span, the contact force and the pull-out value.

(2) Simulation analysis shows that the maintenance cost and the required man-hours increase as the reliability increases. The growth trend is consistent with the law of diminishing marginal benefit. Specifically, when the maintenance strategy is tilted toward the high reliability target, the maintenance cost and required man-hours consumption show the synergistic evolution characteristics of nonlinear increment.

(3) The performance of NSGA-III is compared with various other optimization algorithms in solving the multi-objective optimal maintenance model for contact networks. The results show that NSGA-III has the best optimization results, which further verifies its advantages in solving problems with three or more optimization objectives.

In this study, to reduce the model complexity, a fixed maintenance cycle is used in the modeling process, while the maintenance cycle in the actual engineering scenarios is not fixed due to the influence of equipment state degradation rate, human resource allocation and other dynamic factors. Future research can incorporate the maintenance cycle into the multi-objective optimization framework, construct a flexible decision model considering time-varying maintenance intervals, and then reveal the coupling mechanism between maintenance frequency and system reliability and economy.

## Supporting information

**S1 File. The values used to build** Fig 3.
(XLSX)

**S2 File. The values used to build** Fig 4.
(ZIP)

**S3 File. The values used to build** Fig 5.
(XLSX)

## Acknowledgment

We thank all reviewers and editors for their constructive suggestions and comments.

## Author contributions

**Conceptualization:** Zhijian Qu.

**Data curation:** Xinyu Liao, Rui Chi.

**Formal analysis:** Xinyu Liao.

**Funding acquisition:** Zhijian Qu.

**Investigation:** Xinyu Liao.

**Methodology:** Xinyu Liao, Rui Chi.

**Project administration:** Baoquan Wei, Xinxing Hou.

**Resources:** Xinyu Liao, Rui Chi.

**Software:** Xinyu Liao.

**Supervision:** Rui Chi.

**Validation:** Xinyu Liao.

**Writing – original draft:** Xinyu Liao.

**Writing – review & editing:** Xinyu Liao.

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
