## [Decision Letter · Decision Letter 0]

12 Jan 2025

Dear Dr. Liao,

Thank you for submitting your manuscript to PLOS ONE. After careful consideration, we feel that it has merit but does not fully meet PLOS ONE’s publication criteria as it currently stands. Therefore, we invite you to submit a revised version of the manuscript that addresses the points raised during the review process.

We look forward to receiving your revised manuscript.

Kind regards,

Songpo Yang

Academic Editor

PLOS ONE

Journal Requirements:

“This work was supported by the State Key Laboratory of Performance Monitoring and Protecting of Rail Transit Infrastructure (HJGZ2022203). This research was also supported by the National Natural Science Foundation of China (52462052), Jiangxi Provincial Natural Science Foundation Key Project Fund (20232ACB204025), and Jiangxi Provincial High-level and High-skilled Leading Talents Cultivation Project Fund (202223323).”

“This work was supported by the State Key Laboratory of Performance Monitoring and Protecting of Rail Transit Infrastructure (HJGZ2022203). This research was also supported by the National Natural Science Foundation of China (52462052), Jiangxi Provincial Natural Science Foundation Key Project Fund (20232ACB204025), and Jiangxi Provincial High-level and High-skilled Leading Talents Cultivation Project Fund (202223323).”

 “This work was supported by the State Key Laboratory of Performance Monitoring and Protecting of Rail Transit Infrastructure (HJGZ2022203). This research was also supported by the National Natural Science Foundation of China (52462052), Jiangxi Provincial Natural Science Foundation Key Project Fund (20232ACB204025), and Jiangxi Provincial High-level and High-skilled Leading Talents Cultivation Project Fund (202223323).”

6. Please remove your figures from within your manuscript file, leaving only the individual TIFF/EPS image files, uploaded separately. These will be automatically included in the reviewers’ PDF.

Reviewers' comments:

Reviewer's Responses to Questions

**Comments to the Author**

1. Is the manuscript technically sound, and do the data support the conclusions?

Reviewer #1: Yes

2. Has the statistical analysis been performed appropriately and rigorously?

Reviewer #1: Yes

3. Have the authors made all data underlying the findings in their manuscript fully available?

Reviewer #1: Yes

4. Is the manuscript presented in an intelligible fashion and written in standard English?

Reviewer #1: Yes

Reviewer #1: Abstract: The problem statement is not clear. The findings were not clarified. Include some values (numbers) to expose the results.

Introduction: Capture the research gap and expose the importance of the study.

Conclusions: Rewrite the conclusions in bullet points and expose what was extracted from behind the results. Do not repeat the results.

**Do you want your identity to be public for this peer review?** For information about this choice, including consent withdrawal, please see our Privacy Policy

Reviewer #1: **Yes: ** Prof. Dr. Ahmed Mancy Mosa

---

## [Author Response · Author response to Decision Letter 1]

25 Feb 2025

Manuscript ID: PONE-D-24-52438

Dear Editor,

Thanks a lot for the careful review and instructive comments by the reviewers. In accordance with the comments of editor and reviewer, we have carefully and meticulously revised the manuscript, “Multi-objective maintenance optimization of contact networks considering integrated state parameters”. At the same time, we have addressed the comments of editor and reviewer, and provided our responses as below.

To Editor responses to questions:

To Editor respected Dr. Songpo Yang:

Response: Thank you for pointing out the problems in the manuscript, the authors have corrected the formatting as required by the journal.

2. Please note that PLOS ONE has specific guidelines on code sharing for submissions in which author-generated code underpins the findings in the manuscript. In these cases, we expect all author-generated code to be made available without restrictions upon publication of the work.

Response: Thanks for your comment, the author has uploaded the code and instructions for running it in Github, please search: “Multi-objective-maintenance-optimization-of-contact-networks-considering-integrated-state-parameters” in github.

3. Please provide an amended statement that declares *all* the funding or sources of support (whether external or internal to your organization) received during this study.

Response: Thanks for your comment, all sources of funding or support received during this study are listed below:

“This work was supported by the State Key Laboratory of Performance Monitoring and Protecting of Rail Transit Infrastructure (HJGZ2022203). This research was also supported by the National Natural Science Foundation of China (52462052), Jiangxi Provincial Natural Science Foundation Key Project Fund (20232ACB204025), and Jiangxi Provincial High-level and High-skilled Leading Talents Cultivation Project Fund (202223323). There was no additional external funding received for this study.The funders had no role in study design, data collection and analysis, decision to publish, or preparation of the manuscript.”

4. PLOS requires an ORCID iD for the corresponding author in Editorial Manager on papers submitted after December 6th, 2016. Please ensure that you have an ORCID iD and that it is validated in Editorial Manager.

Response: Thanks for your comment. The author has applied for an ORCID and has updated the information in the “Upload my information”.

5. Please include your amended statements within your cover letter.

Response: Thanks for your comment, we have removed funding-related text from the manuscript, the updated funding statement was given in the third reply to you, please refer to it.

6. Please remove your figures from within your manuscript file, leaving only the individual TIFF/EPS image files.

Response: Thanks for your comment, we have removed all figures from the article.

7. Please review your reference list to ensure that it is complete and correct.

Response: Thanks for your comment. The authors have revised all references in NLM format as required by the journal, and non-English titles have been translated. We have included links to references wherever possible, but some are not DOI-identified and cannot be given. All the cited papers are searchable and there are no cases of retraction.

To Reviewer responses to questions:

To Reviewer respected professor Dr. Ahmed Mancy Mosa:

1. Abstract: The problem statement is not clear. The findings were not clarified. Include some values (numbers) to expose the results.

Response: Sincere thanks to Prof. for his valuable comments. The new manuscript has enriched the presentation of problem and findings in the abstract, and added some results described in numbers. (Bolded text is the part that differs from the original text):

“With the gradual expansion of China's high-speed railroad operation scale and the accumulation of operation time, the pressure of operation and maintenance of contact networks is increasing, and the operation and maintenance strategy of railroad contact networks under the traditional cyclic maintenance mode has the double dilemma of low maintenance efficiency and high labor intensity of maintenance personnel. In order to protect the transportation quality and efficiency of high-speed railroads and reduce the labor intensity of maintenance personnel, this paper proposes a multi-objective optimization operation and maintenance strategy for contact networks considering integrated state parameters, which is able to ensure the high reliability of contact networks while reducing the cost of maintenance and reducing the man-hours required for maintenance. Firstly, according to the integrated state parameters such as the guide height, contact force, pull-out value, and remaining useful life (RUL) of contact networks, the state of contact networks near specific towers is analyzed, revealing the initial reliability of the main equipment including contact wires, positioners, hanging strings, etc., which is used as a parameter of the multi-objective optimal maintenance model. Then, a multi-objective optimal maintenance model is established to achieve the optimization goal of high average reliability of contact networks, low maintenance cost and short required working hours. Finally, the model is solved by the multi-objective optimization algorithm to obtain a series of maintenance solutions in the form of Pareto non-inferior solution sets. Simulation calculations with data from a railroad bureau in southern China yielded that the average reliability of the contact networks system was 0.969, and the maintenance cost was 1,029 thousand yuan, spending 71 man-hours, when the maintenance method generated by the operation and maintenance strategy mentioned in this paper was used.”

2. Introduction: Capture the research gap and expose the importance of the study.

Response: Thank you for pointing out the problems with the manuscript. In the introduction section of the manuscript, we have added a description of current research gaps in the field of contact networks operation and maintenance optimization, highlighted the importance and significance of this study:

Describe the importance and significance of this study:

“……Therefore, the study of a multi-objective optimized maintenance strategy for contact networks based on integrated state parameters detection, and the arrangement of maintenance plan according to the actual state of the contact networks can effectively improve the reliability of the contact network system, reduce the maintenance cost, reduce the man-hours required for the maintenance, improve the labor productivity, and realize the accurate maintenance.……”

“……This method ensures the reliability of system power supply, realizes the precise configuration and dynamic optimization of maintenance resources, effectively avoids the safety risk of “under-maintenance” and the waste of resources of “over-maintenance” that exist in the traditional maintenance mode, improves labor productivity, and provides key technical support for the construction of intelligent operation and maintenance system of contact networks.……”

Describe gaps in existing research:

“……In summary, there are the following gaps in past research on contact networks maintenance optimization: (1) past research usually only focuses on single-objective or bi-objective optimization for reliability and maintenance cost, while the multi-objective optimization research on maintenance hours is in a blank state; (2) past research mainly focuses on how to predict the remaining service life, but research on how to effectively use this information to optimize the contact network maintenance program is rare; (3) the default of past research is to design maintenance programs for contact networks with an initial reliability of 1, i.e., they cannot be applied to contact networks that have just been put into service. The default initial reliability of contact networks is 1, i.e., the maintenance programs are designed for contact networks that have just been put into service, and cannot be applied to contact networks that have been put into service.……”

3. Conclusions: Rewrite the conclusions in bullet points and expose what was extracted from behind the results. Do not repeat the results.

Response: Thank you for pointing out the problems with the manuscript. We have rewritten the conclusion, summarized the research ideas, conclusions obtained and future research directions of this paper. The description in the manuscript is as follows:

“In order to improve the maintenance efficiency of contact networks and reduce the labor intensity of maintenance personnel, this paper proposes a multi-objective optimization operation and maintenance strategy for contact networks considering integrated state parameters. Firstly, the integrated state parameters of contact networks are analyzed to obtain the initial reliability of each main equipment of contact networks. Then, a multi-objective optimization maintenance model is established with the average reliability of contact networks, maintenance cost and required man-hours as the optimization objectives. Finally, the multi-objective optimization algorithm is used to solve the model and obtain a series of maintenance programs. The results show that this method arranges the maintenance plan according to the actual state of the equipment, which can improve the reliability of the contact networks, reduce the maintenance cost, reduce the maintenance man-hours, and provide effective guidance for the operation and maintenance work of the contact networks in the field. In this paper, the following conclusions are obtained in the analysis process:

(1) By analyzing the contact networks 1C detection defect information data of a railroad bureau in southern China, it is concluded that the status of contact networks hanging strings and locators is related to the guide height, the height difference within the span, the contact force and the pull-out value.

(2) Simulation analysis shows that the maintenance cost and the required man-hours increase as the reliability increases. The growth trend is consistent with the law of diminishing marginal benefit. Specifically, when the maintenance strategy is tilted toward the high reliability target, the maintenance cost and required man-hours consumption show the synergistic evolution characteristics of nonlinear increment.

(3) The performance of NSGA-III is compared with various other optimization algorithms in solving the multi-objective optimal maintenance model for contact networks. The results show that NSGA-III has the best optimization results, which further verifies its advantages in solving problems with three or more optimization objectives.

In this study, to reduce the model complexity, a fixed maintenance cycle is used in the modeling process, while the maintenance cycle in the actual engineering scenarios is not fixed due to the influence of equipment state degradation rate, human resource allocation and other dynamic factors. Future research can incorporate the maintenance cycle into the multi-objective optimization framework, construct a flexible decision model considering time-varying maintenance intervals, and then reveal the coupling mechanism between maintenance frequency and system reliability and economy.”

---

## [Decision Letter · Decision Letter 1]

5 Mar 2025

Multi-objective maintenance optimization of contact networks considering integrated state parameters

PONE-D-24-52438R1

Dear Dr. Liao,

We’re pleased to inform you that your manuscript has been judged scientifically suitable for publication and will be formally accepted for publication once it meets all outstanding technical requirements.

Kind regards,

Songpo Yang

Academic Editor

PLOS ONE

Additional Editor Comments (optional):

1、Each of subtitles should be revised in less words for corresponding to there contents.

2、Tables should be shown in piper diagram.

3、Some figures shouldb be updated by clear one.

Reviewers' comments:

Reviewer's Responses to Questions

**Comments to the Author**

Reviewer #1: All comments have been addressed

2. Is the manuscript technically sound, and do the data support the conclusions?

Reviewer #1: Yes

3. Has the statistical analysis been performed appropriately and rigorously?

Reviewer #1: Yes

4. Have the authors made all data underlying the findings in their manuscript fully available?

Reviewer #1: Yes

5. Is the manuscript presented in an intelligible fashion and written in standard English?

Reviewer #1: Yes

Reviewer #1: Overall, I am satisfied with the manuscript in its present form. Therefore, the manuscript can be published in PLOS one.

**Do you want your identity to be public for this peer review?** For information about this choice, including consent withdrawal, please see our Privacy Policy

Reviewer #1: **Yes: ** Prof. Dr. Ahmed Mancy Mosa

---

## [Editor Report · Acceptance letter]

PONE-D-24-52438R1

PLOS ONE

Dear Dr. Liao,

I'm pleased to inform you that your manuscript has been deemed suitable for publication in PLOS ONE. Congratulations! Your manuscript is now being handed over to our production team.

Kind regards,

on behalf of

Dr. Songpo Yang

Academic Editor

PLOS ONE